# A validated protocol to UV-inactivate SARS-CoV-2 and herpesvirus-infected cells

**Timothy K. Soh**[1,2,3,4]**, Susanne Pfefferle**[5,6]**, Stephanie Wurr**[5,7]**, Ronald von Possel**[5,8]**, Lisa Oestereich**[5,7]**, Toni Rieger**[5]**, Charlotte Uetrecht**[1,4,10,11]**, Maria Rosenthal**[1,5,9]***, Jens B. Bosse**[1,2,3,4]*

**1** Centre for Structural Systems Biology, Hamburg, Germany, **2** Hannover Medical School, Institute of Virology, Hannover, Germany, **3** Cluster of Excellence RESIST (EXC 2155), Hannover Medical School, Hannover, Germany, **4** Leibniz Institute of Virology (LIV), Hamburg, Germany, **5** Department of Virology, Bernhard-Nocht Institute for Tropical Medicine, Hamburg, Germany, **6** Virology and Hygiene, Institute for Medical Microbiology, University Center Hamburg-Eppendorf (UKE), Hamburg, Germany, **7** DZIF German Center for Infection Research, Partner Site Hamburg-Lübeck-Borstel-Riems, Hamburg, Germany, **8** Department of Tropical Medicine and Infectious Diseases, Center for Internal Medicine, University of Rostock, Rostock, Germany, **9** Fraunhofer Institute for Translational Medicine and Pharmacology (ITMP), Discovery Research ScreeningPort, Hamburg, Germany, **10** Deutsches Elektronen-Synchrotron (DESY), Hamburg, Germany, **11** Department of Health Sciences and Biomedicine, School of Life Sciences, University of Siegen, Germany

\* jens.bosse@cssb-hamburg.de (JBB); rosenthal@bnitm.de (MR)

**Data Availability Statement:** All relevant data are within the paper.

**Funding:** JBB is funded by the Deutsche Forschungsgemeinschaft (DFG, German Research

## Abstract

Downstream analysis of virus-infected cell samples, such as reverse transcription polymerase chain reaction (RT PCR) or mass spectrometry, often needs to be performed at lower biosafety levels than their actual cultivation, and thus the samples require inactivation before they can be transferred. Common inactivation methods involve chemical crosslinking with formaldehyde or denaturing samples with strong detergents, such as sodium dodecyl sulfate. However, these protocols destroy the protein quaternary structure and prevent the analysis of protein complexes, albeit through different chemical mechanisms. This often leads to studies being performed in over-expression or surrogate model systems. To address this problem, we generated a protocol that achieves the inactivation of infected cells through ultraviolet (UV) irradiation. UV irradiation damages viral genomes and crosslinks nucleic acids to proteins but leaves the overall structure of protein complexes mostly intact. Protein analysis can then be performed from intact cells without biosafety containment. While UV treatment protocols have been established to inactivate viral solutions, a protocol was missing to inactivate crude infected cell lysates, which heavily absorb light. In this work, we develop and validate a UV inactivation protocol for SARS-CoV-2, HSV-1, and HCMV-infected cells. A fluence of 10,000 mJ/cm$^2$ with intermittent mixing was sufficient to completely inactivate infected cells, as demonstrated by the absence of viral replication even after three sequential passages of cells inoculated with the treated material. The herein described protocol should serve as a reference for inactivating cells infected with these or similar viruses and allow for the analysis of protein quaternary structure from *bona fide* infected cells.

Foundation, under Germany's Excellence Strategy – EXC 2155 – project number 390874280 and GRK 2771 project number 453548970 as well as by the Wellcome Trust (https://wellcome.org/) through a Collaborative Award (209250/Z/17/Z).
MR is funded by the Bundesministerium für Bildung und Forschung (BMBF, Federal Ministry of Education and Research, https://www.bmbf.de/) (grant 01KI2019).
CU acknowledges funding from BMG Rapid Response, EU Horizon 2020 ERC StG-2017 759661, BMBF RTK Struktur 01KI20391 grants. The funders did not play any role in the study design, data collection, analysis, preparation of the manuscript, or publication.

**Competing interests:** The authors have declared that no competing interests exist.

## Introduction

Pathogens often require biosafety containment measures during cultivation, but many downstream applications are usually done at lower biosafety levels. Therefore, a validated inactivation protocol is needed, which ideally does not disturb the sample. Aldehyde or solvent fixation is used for microscopy samples while chemical inactivation is frequently employed prior to nucleic acid or protein extractions. The used buffers classically contain compounds such as sodium dodecyl sulfate (SDS) or guanidine thiocyanate [1], which denature the sample and do not keep cells intact. Since they perturb the native protein complexes, they are not suitable for native mass spectrometry (nMS) or protein immunoprecipitation assays without biocontainment measures. In contrast, ultraviolet (UV) radiation mostly crosslinks nucleic acids and closely interacting proteins, which renders them non-functional as polymerase templates [2]. This leads to intact cells that are non-infectious. Importantly, it has been shown that UV irradiation of influenza A reduces the infectious titre and intracellular RNA accumulation but does not affect the hemagglutination titre [3]. This is consistent with damage to the genome but not protein function. Furthermore, UV-inactivated SARS-CoV was able to elicit an antibody response against the structural proteins S and N [4]. In addition, UV inactivation efficiency is wavelength-dependent with shorter wavelengths being more effective [3, 5, 6] and 254 nm being frequently used [5, 7, 8]. Unfortunately, few published protocols describe verified conditions to inactivate whole infected cells. In contrast to viral suspensions, whole cells absorb UV radiation, which results in reduced inactivation efficiency.

In this work, we validated an inactivation protocol that allows for the removal of material from a biosafety containment facility. A fluence of 10,000 mJ/cm$^2$ of 254 nm UV light completely deactivated cells infected with severe acute respiratory syndrome coronavirus type 2 (SARS-CoV-2) or the herpesviruses herpes simplex virus 1 (HSV-1) or human cytomegalovirus (HCMV). Of note, mixing of the whole cell suspension between UV doses was necessary for effective inactivation. The described protocol should serve as a reference for studies that focus on the protein complexes of these or similar pathogens.

## Materials and methods

### Cells and viruses

Vero E6 cells (ATCC cat# CRL-1587) were grown in DMEM with 3% FBS, 1% Penicillin/Streptomycin, 1% Glutamine, and 1% non-essential amino acids. Vero B4 cells (DSMZ ACC 33) were grown in DMEM with 10% FBS. HFF-1 cells (ATCC cat# SCRC-1041) were grown in DMEM with 5% FBS and 1 ng/mL fibroblast growth factor (Pepro Tech EC GmbH, # 100-18B). SARS-CoV-2 (strain human/DEU/HH-1/2020) was amplified in Vero E6 cells. Bacterial artificial chromosome (BAC)-derived HSV-1 strain 17$^+$ was a kind gift from Beate Sodeik [9, 10] and was amplified in Vero B4 cells. HCMV TB40 BAC4 [11] was a kind gift from Wolfram Brune and was amplified in HFF-1 cells. All cells were grown at 37°C with 5% $CO_2$. This study does not require an ethics statement.

### Titration by plaque or FFU assay

Samples were serially diluted in either serum-free DMEM for SARS-CoV-2 or the corresponding growth media for HSV-1 and HCMV. Dilutions started at undiluted, and 200 μL was used to inoculate a monolayer of the corresponding cells in a 24-well format. This lead to a limit of detection (LoD) of 5 PFU/mL. For establishment of irradiation conditions, 200 μL of a 1:100

dilution was the lowest dilution used due to the smaller sample volume, and this generated an LoD of $5x10^2$ PFU/mL. Plates were shaken every 10 min during the 1 h inoculation at 37˚C with 5% $CO_2$. An overlay (0.6% methylcellulose 4000 cP in DMEM with 2% FBS DMEM) was added, and the plates were incubated for 5 days in the case of SARS-CoV-2 and HSV-1 or for 18 days in the case of HCMV. Cells were fixed in 4% formaldehyde in PBS for 30 min. SARS-CoV-2 and HSV-1 plates were stained with 0.5% crystal violet in 10% ethanol, and plaques were counted. HCMV plates were permeabilized with 0.1% Triton X-100 in PBS, stained with anti-IE1/2 hybridoma 3H4 supernatant diluted 1:3 in PBS, and stained with an anti-mouse AF488 antibody (Invitrogen, #A-21202). The 3H4 hybridoma binds to both the IE1 and IE2 proteins from HCMV [12], and it was a kind gift from Thomas Shenk through Wolfram Brune. Fluorescent forming units were counted with an epifluorescence microscope with a standard GFP filter set.

### Detection of infected cells by immunofluorescence

Cells were pelleted at 16,000 xg for 1 min and resuspended in 50 μL PBS. Cells were added to a μ-Slide 18 well (Ibidi, # 81826) for 2 s. Slides were allowed to dry for 30 min at room temperature before inactivation with 100% acetone for 30 min at room temperature. SARS2-CoV-2 samples were blocked with 1% BSA in PBS, stained with anti-N hybridoma 5D4 supernatant diluted 1:10 in 1% BSA in PBS, and stained with anti-mouse AF488 diluted 1:500 and DAPI at 1 ug/mL. The SARS2-CoV-2 antibody was a kind gift from Petra Emmerich. PBS with 1% FBS 0.1% Triton X-100 was used to block HSV-1 and HCMV samples and dilute the antibodies. Anti-ICP0 (Santa Cruz Biotechnology, # sc-53070) diluted 1:100 was used for HSV-1, and anti-IE1/2 hybridoma 3H4 supernatant diluted 1:3 was used for HCMV.

Images were acquired with a Nikon Eclipse Ti2 body equipped with a Yokogawa CSU-W1 spinning disk, Andor iXon Ultra DU-888U3 EMCCD camera, and Plan Apo 20x objective. Pixel resolution was 655 nm/pixel. Illumination was performed with a 405 nm and 488 nm laser through a quad filter (405/488/568/647), and emission light was acquired with a 447/60 and 525/50 filter for DAPI and AF488, respectively.

### UV inactivation

The protocol described in this peer-reviewed article is published on protocols.io, https://doi.org/10.17504/protocols.io.81wgb676qlpk/v1 and is included for printing as S1 File with this article.

The specified cells were seeded into 6-well plates. Infections were performed at an MOI of 0.01 for SARS-CoV-2 and 3 for HSV-1 and HCMV with $2x10^6$ Vero E6 cells, $5x10^5$ Vero B4 cells, and $2x10^5$ HFF-1 cells per individual 6-well, respectively. Infected cells were harvested at 3 dpi for SARS-CoV-2, 1 dpi for HSV-1, and 5 dpi for HCMV. Wells were washed 1x with PBS, and then cells were scraped into 1 mL of PBS. Cells were pelleted at 16,000 xg for 1 min at 4˚C and resuspended in 200 μL PBS. Cells were transferred to a CryoELITE Tissue Vial (Wheaton, #W985100). The open vial along with the upside-down lid were placed inside a UVP Crosslinker that generates 254 nm light from ~16 cm above the samples (Analytik Jena, CL-3000). Samples were irradiated with a fluence of 2,500 mJ/cm$^2$. Where specified, vials were removed from the crosslinker and mixed with a P1000 micropipette. In the final protocol, this irradiation with mixing was repeated 3 additional times for a total fluence of 10,000 mJ/cm$^2$.

To validate the effectiveness of UV inactivation, samples were used to treat fresh cells. Prior to UV treatment, 10 μL were removed from the vial and stored at -80˚C. After UV treatment, 10% of the remaining volume was sampled and the remaining 90% was used to infect a T75 flask of the corresponding cells. For SARS-CoV-2, the cells were split 1:10 every 7 days for 21

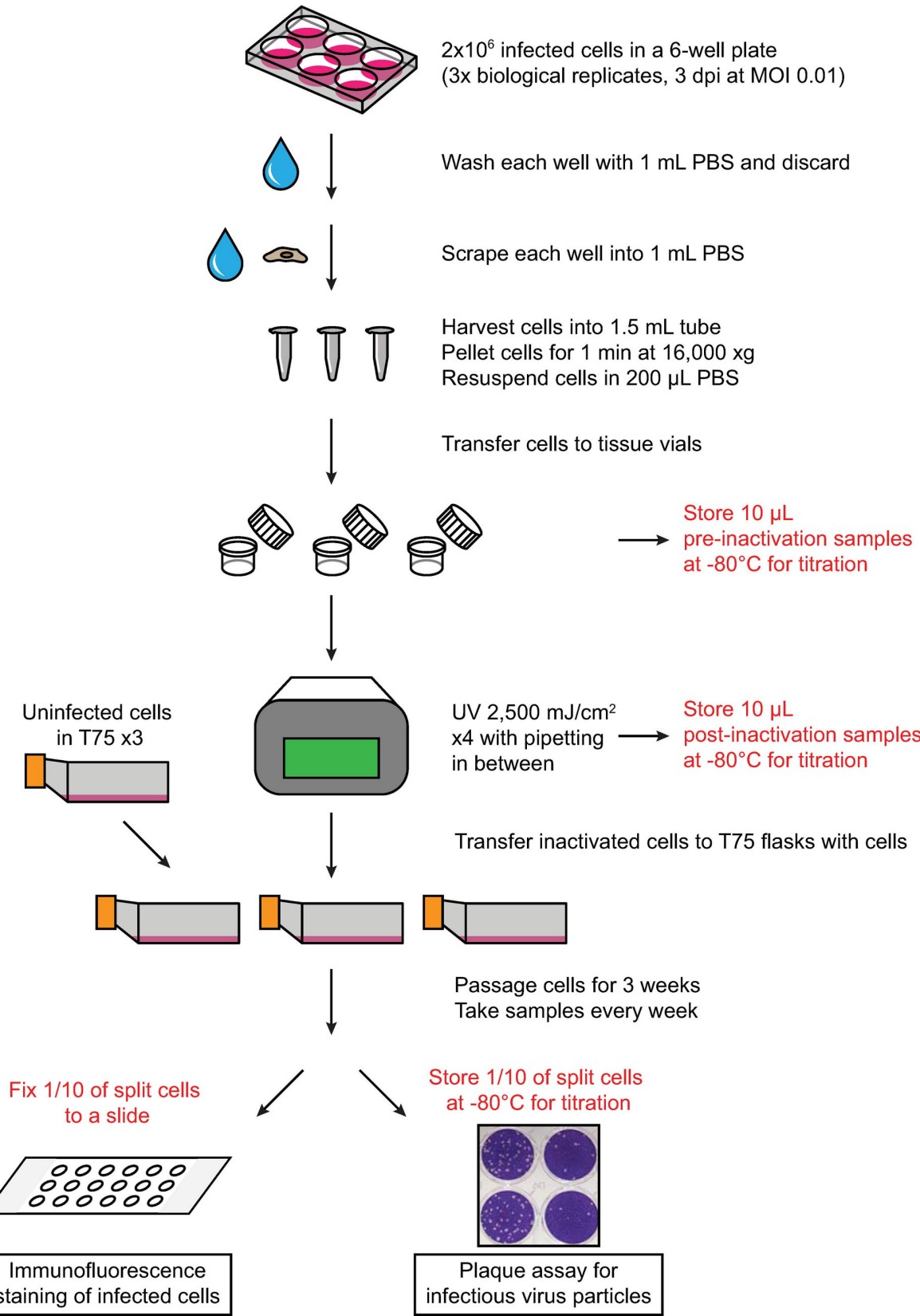

$2x10^6$ infected cells in a 6-well plate
(3x biological replicates, 3 dpi at MOI 0.01)

Wash each well with 1 mL PBS and discard

Scrape each well into 1 mL PBS

Harvest cells into 1.5 mL tube
Pellet cells for 1 min at 16,000 xg
Resuspend cells in 200 μL PBS

Transfer cells to tissue vials

Store 10 μL
pre-inactivation samples
at -80°C for titration

Uninfected cells
in T75 x3

UV 2,500 mJ/cm$^2$
x4 with pipetting
in between

Store 10 μL
post-inactivation samples
at -80°C for titration

Transfer inactivated cells to T75 flasks with cells

Passage cells for 3 weeks
Take samples every week

Fix 1/10 of split cells
to a slide

Store 1/10 of split cells
at -80°C for titration

Immunofluorescence
staining of infected cells

Plaque assay for
infectious virus particles

**Fig 1. Schematic of validation procedure of UV inactivation protocol for SARS-CoV-2.** Samples were generated with 3 biological replicates under the conditions that lead to no detectable virus. Fresh cells were treated with the UV irradiated samples, and these treated cells were passaged for three weeks to test for viral replication. The inactivation protocol is the same for HSV-1 and HCMV, except for differences in the number of cells and passaging times, which are described in the Materials and Methods section.

days. For HSV-1, the cells were split 1:10 every 3–4 days for 21 days. For HCMV, the cells were split 1:2 every 2 weeks for 6 weeks. At the time of splitting, 10% of the cells were saved for titration by plaque assay, and 10% were saved for detection of infected cells by immunofluorescence. For SARS-CoV-2 and HSV-1, samples were taken every 7 days, and for HCMV, samples were taken every 2 weeks. UV-inactivation of samples was performed with 3 biological replicates. The positive control was transferred to the tissue vial but not irradiated. Viable SARS-CoV-2 was exclusively handled in a BSL3 facility at the Bernhard-Nocht Institute for Tropical Medicine (BNITM). Experiments with infectious HSV-1 and HCMV were performed in appropriate BSL2 facilities at the Hannover Medical School (MHH).

## Results

### UV inactivation of virus infected cells

The ability of UV irradiation to inactivate infected cells was tested (Fig 1). A sample was generated from a single 6-well. Depending on the cell type, each sample contained approximately $10^6$ infected cells. The supernatant was removed, and the cells were washed once with PBS. The cells were scraped into PBS, pelleted, and resuspended in 200 µL of PBS. The resuspended cells were transferred to a tissue vial for UV irradiation. These vials were chosen because of the large surface area of the bottom and the ability to tightly close the lid and disinfect the outside surface. The samples were spread out over the entire surface of the vial to maximize the surface

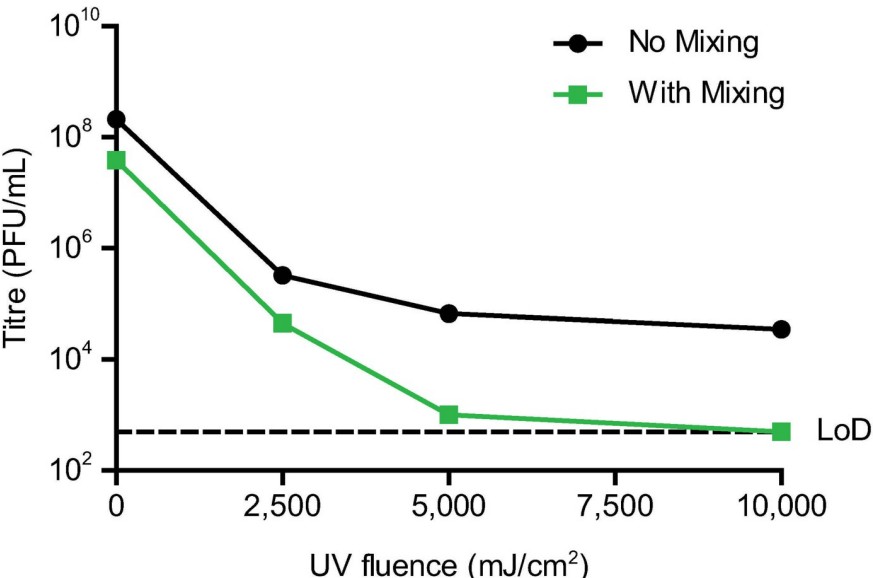

**Fig 2. Mixing cells during irradiation increases inactivation efficiency.** Vero B4 cells were infected with HSV-1 and irradiated in tissue vials. Cells were treated with the indicated irradiation dose in a single interval (No Mixing, black line) or were treated repeatedly with a fluence of 2,500 mJ/cm² followed by mixing for the total dose stated (With Mixing, green line). Sample inactivation was evaluated by plaque assay with a limit of detection (LoD) of 5x10² PFU/ mL. Each dose curve was performed with n = 1.

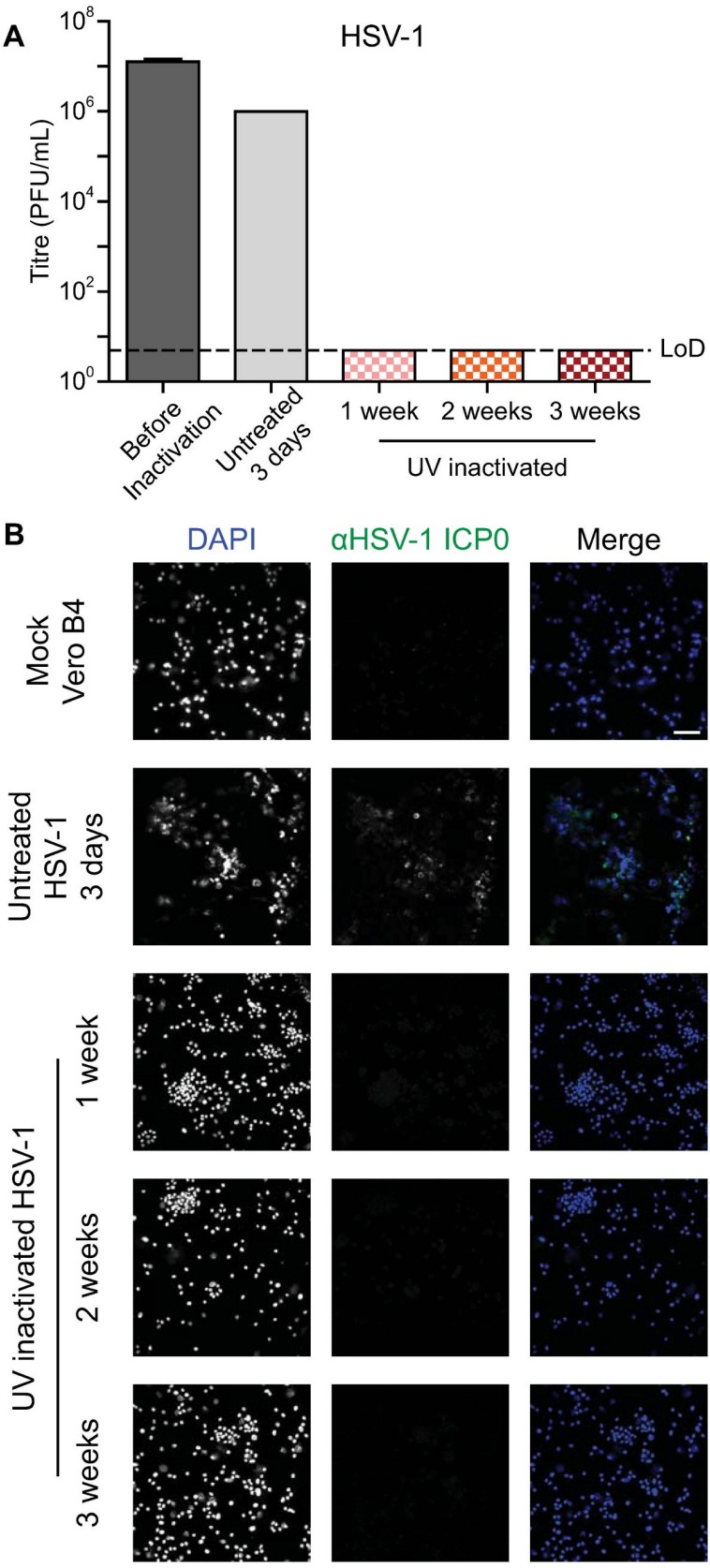

**Fig 3. Inactivation of HSV-1 infected cells.** Vero B4 cells infected with HSV-1 were UV-inactivated. (A) The virus titres of samples taken during inactivation validation were determined with a limit of detection (LoD) of 5 PFU/mL. The before inactivation and UV inactivated samples were performed with biological replicates, n = 3, and the standard deviation is shown. The untreated control sample was performed with n = 1. (B) Immunofluorescence imaging of cells treated with the UV-inactivated infected cells was performed with biological replicates, n = 3. Cells were stained for the HSV-1 protein ICP0 and with DAPI. The scale bar represents 100 μm.

area of the bubble that formed in the middle. The maximal depth was 5 mm. Different fluences were tested, and Fig 1 illustrates the final protocol.

An initial pre-study dose curve to determine the UV dosage was performed under BSL2 conditions with Vero B4 cells infected with HSV-1 (Fig 2). HSV-1 is a fast growing BSL2 organism that allows for quick iterations on the protocol. Cells were treated with an increasing fluence of UV in either a single dose or in intervals of 2,500 mJ/cm$^2$ with mixing in between. After mixing, any remaining bubbles were moved to the edge of the vial. As a single dose, UV treatment was insufficient to completely inactivate the infected cells, and inactivation efficiency appeared to plateau. To overcome the plateau, irradiation with intermittent mixing was performed, and this resulted in the reduction of HSV-1 to below the limit of detection. We demonstrated that the minimal dose required to inactivate HSV-1 is between 5,000 mJ/cm$^2$ and 10,000 mJ/cm$^2$.

Once a sufficient dosage for inactivation of HSV-1 was established under BSL2 conditions, validation was performed and additional viruses were included to test the sufficiency of this dosage. Each of the tested viruses were grown in cell lines supporting viral growth to high titers. Vero B4 cells were infected with HSV-1 (Fig 3), HFF-1 cells were infected with HCMV (Fig 4), and Vero E6 cells were infected with SARS-CoV-2 (Fig 5). Cells were UV-inactivated and then added to fresh cells to validate the inactivation method. The treated cells were passaged, and samples were taken at regular intervals. The samples were evaluated for infection by immunofluorescence and infectious virus by titration. The multi-week passaging protocol was designed in accordance with in-house validation regulations.

We found that 10,000 mJ/cm$^2$ with sample mixing is sufficient for complete inactivation of three viruses: HSV-1, HCMV, and SARS-CoV-2. This dose was optimized with HSV-1, and the necessary dose was found to be in between 5,000 and 10,000 mJ/cm$^2$. Previous work found that cell-free HSV-1 requires 50 mJ/cm$^2$ for a $10^4$ fold reduction in titre [13]; in contrast, we found that >2,500 mJ/cm$^2$ was necessary for a $10^4$ fold reduction in titre of infected cells. While this could be due to differences in experimental setup, the cells are likely absorbing the UV light and blocking the exposure of all cells at once, and mixing provided a means to facilitate more uniform irradiation.

## Discussion

Here we report a protocol to UV-inactivate whole virus-infected cells that could serve as a reference for other viruses and researchers as in-house optimization and validation is a time-consuming process. Previous studies have focused on UV treatment of viral supernatants [3, 5, 7, 8] with the objective to inactivate soluble viral particles [4]. The fluence required to inactivate clarified viral supernatant varies between publications. A fluence of 1,446 mJ/cm$^2$ was required for complete inactivation of SARS-CoV supernatant [7] while a fluence of 40 mJ/cm$^2$ [8] or 1,048 mJ/cm$^2$ [5] was required for SARS-CoV-2 supernatants. This variation could be due to the volume depth since water and medium solutes absorb UV light. Consistent with this, the lowest fluence was required with the smallest volume [8]. An additional factor could be the distance to the UV light source. While the fluence would partially incorporate this variable, the angle of illumination could affect the uniformity of irradiation and blockage from the vessel

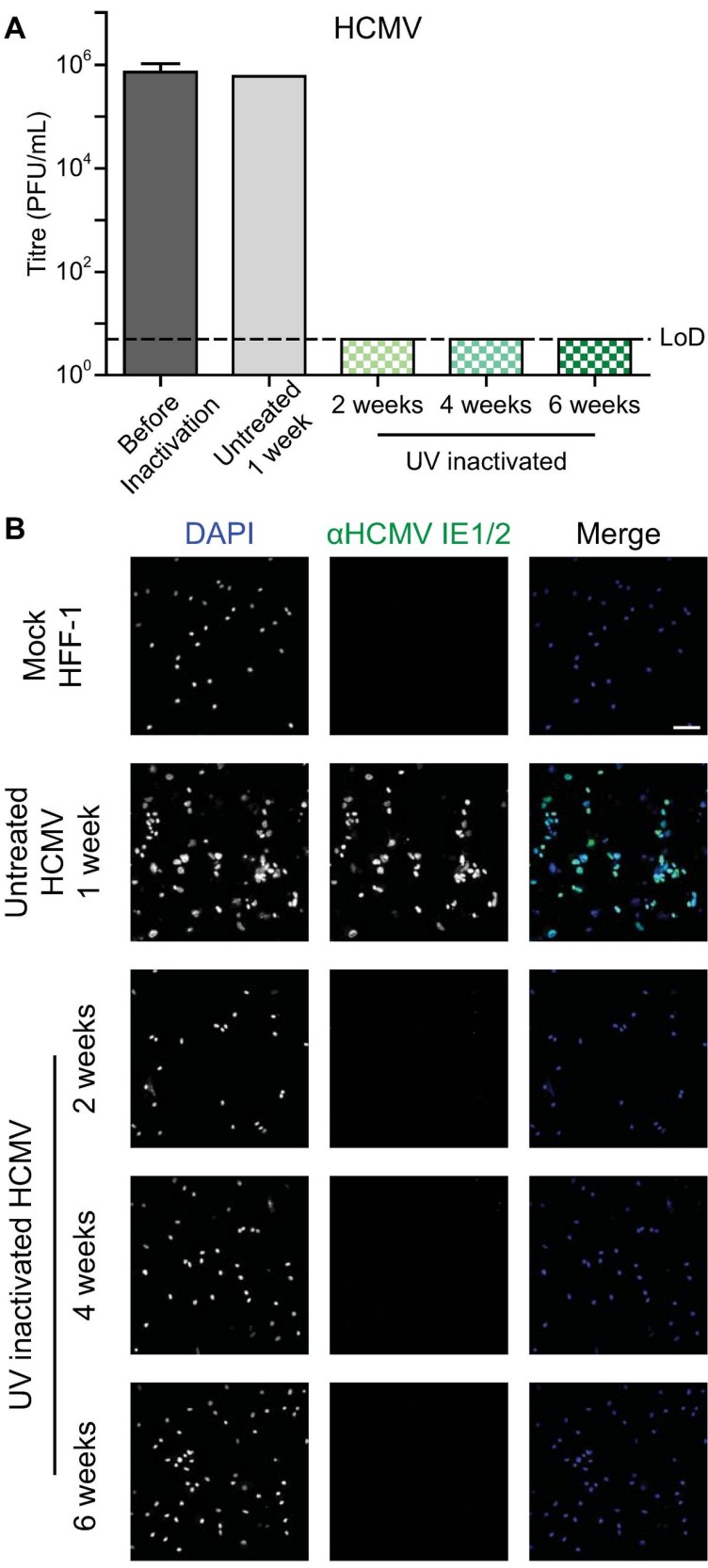

**Fig 4. Inactivation of HCMV infected cells.** HFF-1 cells infected with HCMV were UV-inactivated. (A) The virus titres of samples taken during inactivation validation were determined with a limit of detection (LoD) of 5 PFU/mL. The before inactivation and UV inactivated samples were performed with biological replicates, n = 3, and the standard deviation is shown. The untreated control sample was performed with n = 1. (B) Immunofluorescence imaging of cells treated with the UV-inactivated infected cells was performed with biological replicates, n = 3. Cells were stained for the HCMV proteins IE1/2 and with DAPI. The scale bar represents 100 μm.

walls. The work presented here demonstrates the sufficiency of 10,000 mJ/cm$^2$ at a height of 16 cm to inactivate approximately 10$^6$ intact cells resuspended in 200 μL of PBS. However, this amount of cells might not be sufficient for certain downstream assays, such as mass spectrometry. In this case, multiple wells can be pooled post-inactivation, but the concentration of cells should not be changed as the efficiency of UV inactivation is likely sensitive to the optical density of the sample solution. Organoid cultures are becoming popular model systems since they are more similar to *in vivo* conditions than monolayers of immortalized cells [14]. The herein described conditions may not be valid with these larger and optically denser cellular structures. In addition, samples such as serum, plasma, stool, saliva, or tissue often have pigments that would likely interfere with UV treatment. However, for standard cell culture models, the presented UV inactivation protocol is effective.

The dosage used for all three viruses was based on the dose curve for HSV-1. While herpesviruses are double-stranded DNA (dsDNA) viruses and coronaviruses are single-stranded RNA (ssRNA) viruses, it was assumed that a similar dose would be required for complete inactivation, namely a 10$^6$ fold reduction in titre. The sensitivity of viruses to UV has been reported to be dependent on whether the genome is DNA or RNA as well as whether it is single- or double-stranded. However, this rule is not universally true. One study compared a single virus from each group and found that the DNA viruses required about twice as much 254 nm UV irradiation to achieve the same reduction in titre as the corresponding RNA virus [15]. In addition, the double-stranded viruses required about triple the UV dose as the corresponding single-stranded virus to be equally inactivated. Another study compared 11 viruses and similarly found that on average, a 2–3 fold higher dose is required for DNA and double-stranded viruses [16]. Although, there are exceptions to this pattern even among the small number of viruses that were tested. Reports in the literature show high variability within each viral group. HSV-1 requires 50 mJ/cm$^2$ for a 4-log inactivation [13] while adenovirus, another dsDNA virus, requires 180 mJ/cm$^2$ [17]. Given that every publication uses a different experimental setup, it is difficult to interpret whether these differences in dose are due to an intrinsic or experimental difference. It is unclear whether a difference in sensitivity between DNA and RNA is an intrinsic property. Purified dsDNA requires 1 mJ/cm$^2$ to damage 50% of the strands [18] while purified ssRNA requires 3 mJ/cm$^2$ [19]. Since the assay for UV inactivation is replication in the target cell, it is possible that nuclear replicating DNA viruses benefit from cellular DNA repair machinery. However, in order to provide an opportunity for viral gene expression, repair of the incoming genome would need to conclude before activation of the innate immune system. For HSV-1, incoming genomes associate with innate immune proteins in 15–30 min [20]. In comparison, 50% of cellular UV damage is repaired in 2 h [18]. These kinetics make it unlikely for DNA damage repair mechanisms to play a major role in rescuing UV-damaged viruses. The sensitivity of a virus to UV is not definitively based on its genome type. The precise minimal dose would need to be determined for a given virus, but the dosage developed in this work is sufficient for a dsDNA and ssRNA virus.

Since UV inactivation crosslinks nucleic acids and associated proteins, it can be used to study their interaction [21]. The VP30 protein from ebola virus was shown to bind to the leader region of the genome [22], and the bovine coronavirus N protein was demonstrated to

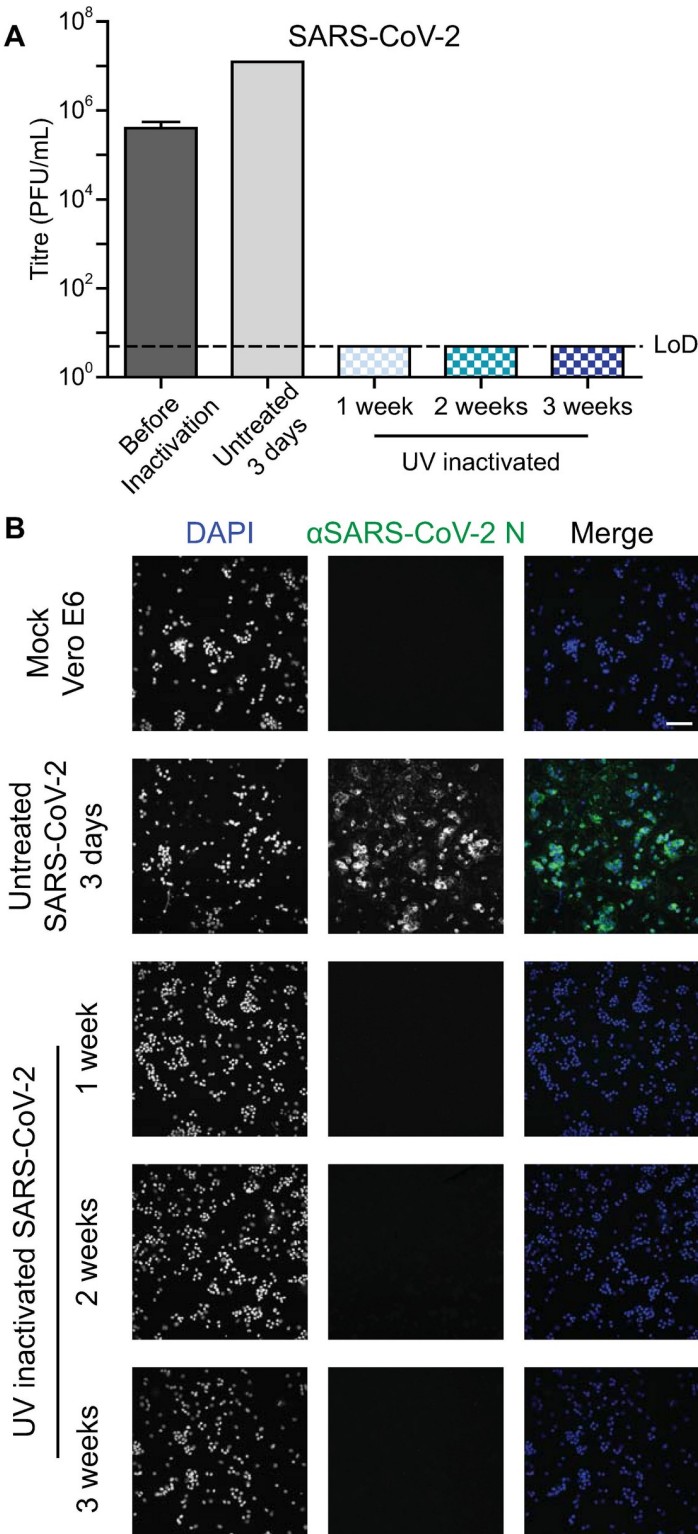

**Fig 5. Inactivation of SARS-CoV-2 infected cells.** Vero E6 cells infected with SARS-CoV-2 were UV-inactivated. (A) The virus titres of samples taken during inactivation validation were determined with a limit of detection (LoD) of 5 PFU/mL. The before inactivation and UV inactivated samples were performed with biological replicates, n = 3, and the standard deviation is shown. The untreated control sample was performed with n = 1. (B) Immunofluorescence imaging of cells treated with the UV-inactivated infected cells was performed with biological replicates, n = 3. Cells were stained for the SARS-CoV-2 protein N and with DAPI. The scale bar represents 100 μm.

bind to poly(A) tails inside cells [23]. These experiments were performed with 1,800 mJ/cm$^2$ and 1,200 mJ/cm$^2$ of 254 nm UV light, respectively. Additionally, since the effects of UV radiation are non-specific, it can be used to crosslink endogenous RNA, and this was achieved with a dose of only 150–400 mJ/cm$^2$ in cells [24]. This concept has been expanded to the methods of RNA-interactome capture (RIC) and cross-linking and immunoprecipitation (CLIP), which employ UV crosslinking followed by mass spectrometry or sequencing, respectively [21]. Given the much higher dose necessary for inactivation in the presented work, nucleic acid binding proteins will likely be crosslinked to their target. This stabilization of protein interactions could be advantageous in the study of intermolecular interactions.

UV radiation can also crosslink proteins, and this can affect downstream analyses, depending on the readout. Loveday *et al.* [25] found that 75 mJ/cm$^2$ of 254 nm UV light was sufficient to inactivate clarified supernatant of SARS-CoV-2 by 10$^8$-fold. They found that UV irradiation decreased detection by RT-PCR, as expected for crosslinked RNA, starting with a fluence of 14 mJ/cm$^2$. Protein crosslinking could be detected by SDS-PAGE with a UV fluence of 2,100 mJ/cm$^2$, and this fluence sufficiency damaged the viral epitopes to decrease detection by ELISA. In contrast, the ultrastructure measured by negative stain electron microscopy was unperturbed. While UV has the potential to crosslink proteins, the quantitative effect will likely depend on the actual energy absorbed per cell. Since we observed that mixing of the sample is necessary, individual cells will likely experience only a fraction of the total energy applied. Therefore, the effect on protein-protein interactions needs to be evaluated on a case-by-case basis, but crosslinking could be advantageous in preserving virus-host complexes.

We demonstrated that a dose of 10,000 mJ/cm$^2$ was sufficient for complete inactivation of infected cells. The minimal dose to inactivate HSV-1 is in between 5,000 and 10,000 mJ/cm$^2$, and based on publications that compared multiple viruses directly, the dose necessary for other viruses is likely within 3-fold [15, 16]. While this dose is quite high compared to most protocols, it is necessary for at least HSV-1, and undesirable effects associated with this dose are a limitation of this procedure. Since this protocol will mediate the removal of material from high containment facilities, it is preferable to err on the side of excessive inactivation. In summary, this validated inactivation protocol provides a basis for analyzing various virus-infected cell samples and could be used to analyze the quaternary structure of virus-host complexes without the need for biocontainment.

## Supporting information

**S1 File. Protocol.io version of UV inactivation protocol.** Protocol describing UV inactivation of infected cells. Also available at https://doi.org/10.17504/protocols.io.81wgb676qlpk/v1. (PDF)

## Acknowledgments

The HCMV TB40 BAC4 virus was a kind gift from Wolfram Brune, and the 3H4 anti-HCMV IE1/2 antibody was a kind gift from Thomas Shenk through Wolfram Brune. We thank Beate Sodeik for the kind gift of BAC-derived HSV-1 and give thanks for providing the SARS-CoV-2 monoclonal antibody 5D4 to Petra Emmerich (BNITM).

## Author Contributions

**Conceptualization:** Charlotte Uetrecht, Timothy K. Soh, Lisa Oestereich, Toni Rieger, Maria Rosenthal, Jens B. Bosse.

**Data curation:** Timothy K. Soh.

**Formal analysis:** Timothy K. Soh.

**Funding acquisition:** Charlotte Uetrecht, Maria Rosenthal, Jens B. Bosse.

**Investigation:** Timothy K. Soh.

**Methodology:** Charlotte Uetrecht, Timothy K. Soh, Lisa Oestereich, Toni Rieger, Maria Rosenthal.

**Project administration:** Charlotte Uetrecht, Maria Rosenthal, Jens B. Bosse.

**Resources:** Charlotte Uetrecht, Susanne Pfefferle, Stephanie Wurr, Ronald von Possel, Maria Rosenthal.

**Supervision:** Lisa Oestereich, Maria Rosenthal, Jens B. Bosse.

**Validation:** Timothy K. Soh.

**Visualization:** Timothy K. Soh.

**Writing – original draft:** Timothy K. Soh, Jens B. Bosse.

**Writing – review & editing:** Timothy K. Soh, Susanne Pfefferle, Stephanie Wurr, Ronald von Possel, Lisa Oestereich, Toni Rieger, Maria Rosenthal, Jens B. Bosse.

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
