## [Decision Letter · Decision Letter 0]

9 Aug 2022

PONE-D-22-19938A validated protocol to UV-inactivate SARS-CoV-2 and herpesvirus-infected cellsPLOS ONE

Dear Dr. Bosse,

Thank you for submitting your manuscript to PLOS ONE. After careful consideration, we feel that it has merit but does not fully meet PLOS ONE’s publication criteria as it currently stands. Therefore, we invite you to submit a revised version of the manuscript that addresses the points raised during the review process. The comments raised by the reviewers include the reduction of UV dose. Please discuss this carefully. 

We look forward to receiving your revised manuscript.

Kind regards,

Etsuro Ito

Academic Editor

PLOS ONE

Journal Requirements:

"This work was funded by the Deutsche Forschungsgemeinschaft (DFG, German Research Foundation) under Germany's Excellence Strategy – EXC 2155 – project number 390874280 as well as by the Wellcome Trust through a Collaborative Award (209250/Z/17/Z) to JBB. We acknowledge support by the German Federal Ministry for Education and Research (grant 01KI2019) to MR. The HCMV TB40 BAC4 virus was a kind gift from Wolfram Brune, and the 3H4 anti HCMV IE1/2 antibody was a kind gift from Thomas Shenk through Wolfram Brune. We thank Beate Sodeik for the kind gift of BAC-derived HSV-1. Thanks for providing the SARS-CoV-2 monoclonal antibody 5D4 to Petra Emmerich (BNITM)"

"JBB is funded by the Deutsche Forschungsgemeinschaft (DFG, German Research Foundation, https://www.dfg.de/) under Germany's Excellence Strategy – EXC 2155 – project number 390874280 as well as by the Wellcome Trust (https://wellcome.org/) through a Collaborative Award (209250/Z/17/Z).

MR is funded by the Bundesministerium für Bildung und Forschung (BMBF, Federal Ministry of Education and Research, https://www.bmbf.de/) (grant 01KI2019).

The funders did not play any role in the study design, data collection, analysis, preparation of the manuscript, or publication."

4. To comply with PLOS ONE submissions requirements, please provide the Protocols.io DOI in the Methods section of the manuscript using this format: “The protocol described in this peer-reviewed article is published on protocols.io, https://dx.doi.org/10.17504/protocols.io[........] and is included for printing as supporting information file 1 with this article.” Please also provide the Protocols.io DOI in the “Protocol DOI” field of the submission form (via “Edit Submission”). For more information, please see our submission guidelines:  https://journals.plos.org/plosone/s/submission-guidelines#loc-guidelines-for-specific-study-types

5. We note you have not provided a Protocol.io PDF version of your protocol. As noted in our submission requirements, please upload a Protocol.io PDF version of your protocol as a Supporting Information file and name the file ‘S1 file’. Please update your Supporting Information Captions if necessary. If you have not yet uploaded your protocol to Protocols.io you are welcome to use the Protocols.io customer service code ‘PLOS2021.’ When using this customer code while submitting to Protocols.io, please make reference to your PLOS ONE submission, including your PLOS ONE manuscript number. With this customer code, Protocols.io editorial staff will import and format your protocol at no charge. For more information, please see our submission guidelines:  https://journals.plos.org/plosone/s/submission-guidelines#loc-guidelines-for-specific-study-types

Reviewers' comments:

Reviewer's Responses to Questions

**Comments to the Author**

1. Does the manuscript report a protocol which is of utility to the research community and adds value to the published literature?

Reviewer #1: Yes

Reviewer #2: Yes

2. Has the protocol been described in sufficient detail?

Descriptions of methods and reagents contained in the step-by-step protocol should be reported in sufficient detail for another researcher to reproduce all experiments and analyses. The protocol should describe the appropriate controls, sample sizes and replication needed to ensure that the data are robust and reproducible.

Reviewer #1: Partly

Reviewer #2: Yes

3. Does the protocol describe a validated method?

Reviewer #1: Yes

Reviewer #2: Yes

4. If the manuscript contains new data, have the authors made this data fully available?

Reviewer #1: N/A

Reviewer #2: Yes

**5. Is the article presented in an intelligible fashion and written in standard English?**

Reviewer #1: Yes

Reviewer #2: Yes

6. Review Comments to the Author

Reviewer #1: In the submitted manuscript, authors seek to validate a UV inactivation protocol for virally infected cells. UV exposure of virus suspensions is very effective for the stabilization of protein conformations and activity while rendering highly pathogenic viruses incapable of propagating infection. There are many limitations to UV treatment, with particular regards sample complexity. Evaluation of UV conditions to inactivate virus within cells has not been previously performed. Authors compare the effectiveness of UV exposure for 3 viruses; HSV-1, CMV, and SARS-CoV-2. Using plaque assay and immunofluorescence assays, they demonstrate that 10,000 mJ/cm^2 of UV 257nm irradiation will completely eliminate infectivity from infected cell suspensions of the 3 viruses.

Overall, the study is framed well with the introduction and discussion. The evaluation of residual infectivity with multi-week passage of exposed cells is rigorous. Testing of viral infectivity by both plaque/FFA assay and immunofluorescence detection of infected cells is equally robust. Experimentally, there is a gap regarding the “sufficiency” of 10,000 mJ exposure and the equivalency of inactivation from the 3 viruses. While inactivation conditions are validated for HSV-1, CMV, and SARS-2, it is unclear if this much exposure is necessary or possibly detrimental to downstream processes.

In addition, the discussion needs to be enhanced with a brief comparison of UV inactivation results between RNA and DNA viruses. The results section needs additional information and explanation of experimental rationale and interpretation. There are additional minor details that need to be addressed.

Major Concerns:

1) Equivalency of DNA and RNA viruses to UV inactivation. The sensitivity of viruses to UV irradiation is variably reported in the literature. Authors are correct in identify differences in containers, distance, etc. But equally, DNA and RNA viruses will have different reactions to UV irradiation exposure. DNA viruses have a limited capacity to repair lesions induced by UV crosslinking that RNA viruses don’t necessarily have. UV sensitivity is only tested for HSV-1 in figure 1, while all subsequent experiments are performed at the maximal UV dose. Authors should briefly address the possible discrepancy between nucleic acid types in the discussion. Better would be a comparison of sensitivity to UV inactivation for SARS-CoV-2 infected cells.

2) Line 168 “We found that the UV dose needed to inactivate infected cells was much higher than…” This line is difficult to reconcile. First, evaluation of UV sensitivity of cell-free virus suspensions would be a necessary comparison in your system to declare that infected cells require more. Secondly, for CMV and SARS-2 intermediate UV exposures are not tested to demonstrate that 10,000 mJ is necessary. Emphasizing that you have validated conditions of complete inactivation is all that is proven. Further comparison of relative dosing is not appropriate in the results or discussion.

3) Explanation of results – The results section is lacking in explanation of experimental rationale. Multiple questions can/should be addressed in this section.

a. What is the geometry of the cell suspension in the vial. Vial dimensions are very large (5 mL max volume) while volume is only 200 uL. Did authors attempt other volumes of resuspension? As they note, cell density will influence the optical density which could reduce UV efficacy through absorbance and shading. Was the drop spread out? What was the depth? Did a larger volume change UV inactivation efficiency?

b. Why was a multi-week passaging of potentially infected cells carried out? Was plaque assay/IFA performed from UV inactivated cell suspensions directly? The former is very rigorous but seems somewhat unnecessary.

Minor Concerns:

1) Line 29 – “However, these protocols destroy…” It is unclear why equivalency between formaldehyde cross-linking and sds denaturation are being made here.

2) Line 170 – “This was likely due to cells…” The geometry of the container also needs to be taken into consideration. The top of cell suspension is the area receiving UV irradiation and will rapidly lose efficacy within the solution. What was the depth of solution in the vial?

3) Line 187 “certain downstream assays.” It is unclear at this point in the manuscript what these downstream assays would entail. Subsequent explanation of CLIP and other nucleic acid association assays are reasonable, but disconnected from this statement.

4) Figure 2 title – “Pre-study experiment” is not informative and should be revised. The higher LOD is not sufficiently explained but presumptively reflects the 10 uL sample volume being dilute 1:100 for initial analysis? It is unclear if this "n" refers to the number of replicate experiments or samples evaluated at each UV inactivation dose.

5) Supplemental Instructions line 23 – The last step should indicate the agent of disinfection and time, i.e. 70% ethanol for 5 minutes.

Reviewer #2: Nice work. Please spell out short from in first line of Abstract. UV light can damage proteins. This may be something to keep in mind in future work. The UV dose you are using is quite high. It would be good if you could reduce it.

7. PLOS authors have the option to publish the peer review history of their article (what does this mean?). If published, this will include your full peer review and any attached files.

Reviewer #1: No

Reviewer #2: **Yes: **John Gibson

---

## [Author Response · Author response to Decision Letter 0]

18 Aug 2022

We sincerely thank all reviewers for their critical reading of our manuscript and their insightful and fair suggestions. Below is our response to the reviewers’ comments (highlighted in light blue). References have been added to the reference list. Line numbers refer to the document with tracked changes.

Reviewer #1: In the submitted manuscript, authors seek to validate a UV inactivation protocol for virally infected cells. UV exposure of virus suspensions is very effective for the stabilization of protein conformations and activity while rendering highly pathogenic viruses incapable of propagating infection. There are many limitations to UV treatment, with particular regards sample complexity. Evaluation of UV conditions to inactivate virus within cells has not been previously performed. Authors compare the effectiveness of UV exposure for 3 viruses; HSV-1, CMV, and SARS-CoV-2. Using plaque assay and immunofluorescence assays, they demonstrate that 10,000 mJ/cm^2 of UV 257nm irradiation will completely eliminate infectivity from infected cell suspensions of the 3 viruses.

Overall, the study is framed well with the introduction and discussion. The evaluation of residual infectivity with multi-week passage of exposed cells is rigorous. Testing of viral infectivity by both plaque/FFA assay and immunofluorescence detection of infected cells is equally robust. Experimentally, there is a gap regarding the “sufficiency” of 10,000 mJ exposure and the equivalency of inactivation from the 3 viruses. While inactivation conditions are validated for HSV-1, CMV, and SARS-2, it is unclear if this much exposure is necessary or possibly detrimental to downstream processes.

In addition, the discussion needs to be enhanced with a brief comparison of UV inactivation results between RNA and DNA viruses. The results section needs additional information and explanation of experimental rationale and interpretation. There are additional minor details that need to be addressed.

We revised the manuscript according to the reviewer’s suggestions. We explained our rationale for establishing inactivation conditions using HSV-1 as a baseline (line 174). We also included a discussion of UV inactivation of DNA versus RNA viruses (line 264). The rationale for mixing during irradiation, the choice of tissue vial, and further experimental details have been added to the results section (line 160, 177, 178).

Major Concerns:

1) Equivalency of DNA and RNA viruses to UV inactivation. The sensitivity of viruses to UV irradiation is variably reported in the literature. Authors are correct in identify differences in containers, distance, etc. But equally, DNA and RNA viruses will have different reactions to UV irradiation exposure. DNA viruses have a limited capacity to repair lesions induced by UV crosslinking that RNA viruses don’t necessarily have. UV sensitivity is only tested for HSV-1 in figure 1, while all subsequent experiments are performed at the maximal UV dose. Authors should briefly address the possible discrepancy between nucleic acid types in the discussion. Better would be a comparison of sensitivity to UV inactivation for SARS-CoV-2 infected cells.

We added a section to the discussion where we summarize the literature on UV sensitivity of DNA versus RNA as well as the potential influence of DNA repair mechanisms (line 264).

2) Line 168 “We found that the UV dose needed to inactivate infected cells was much higher than…” This line is difficult to reconcile. First, evaluation of UV sensitivity of cell-free virus suspensions would be a necessary comparison in your system to declare that infected cells require more. Secondly, for CMV and SARS-2 intermediate UV exposures are not tested to demonstrate that 10,000 mJ is necessary. Emphasizing that you have validated conditions of complete inactivation is all that is proven. Further comparison of relative dosing is not appropriate in the results or discussion.

We thank the reviewer for pointing this out. We changed the language to better reflect that we used a sufficient dosage that is safe for all tested viruses but which might not necessarily be the lowest required dosage for inactivation for individual viruses (line 182, 191, 229).

3) Explanation of results – The results section is lacking in explanation of experimental rationale. Multiple questions can/should be addressed in this section.

a. What is the geometry of the cell suspension in the vial. Vial dimensions are very large (5 mL max volume) while volume is only 200 uL. Did authors attempt other volumes of resuspension? As they note, cell density will influence the optical density which could reduce UV efficacy through absorbance and shading. Was the drop spread out? What was the depth? Did a larger volume change UV inactivation efficiency?

We added this information to the results section (line 162). Briefly, the sample was spread out on the bottom of the vial, but given the small volume, it still formed a droplet. This droplet has a maximal depth of 5 mm. During optimization, cells were resuspended in 100 μL to 300 μL prior to UV treatment. There was no clear difference in efficiency. As about 20-40 μL of sample volume is lost during handling due to adherence to the container walls, 200 μL was chosen as a compromise to minimize the volume while reducing the unrecovered fraction.

b. Why was a multi-week passaging of potentially infected cells carried out? Was plaque assay/IFA performed from UV inactivated cell suspensions directly? The former is very rigorous but seems somewhat unnecessary.

The multi-week passaging is required by our institute’s biosafety regulations. We noted this in the results section (line 199). We appreciate the recognition of the rigour of this requirement. A plaque assay/IFA was not performed directly on the UV-treated material since it would be measured after the material had an opportunity to replicate.

Minor Concerns:

1) Line 29 – “However, these protocols destroy…” It is unclear why equivalency between formaldehyde cross-linking and sds denaturation are being made here.

The reviewer is correct. These chemicals function through different mechanisms. We modified the manuscript accordingly to emphasize that both compounds obstruct the identification of physiological protein complex compositions, although through different mechanisms (line 33). High concentrations of formaldehyde that are sufficient to inactivate the sample will crosslink proteins that are not part of the same complex, and reversal of the crosslinker through heat will destroy the protein complex interactions.In contrast, SDS will directly disrupt the protein interactions.

2) Line 170 – “This was likely due to cells…” The geometry of the container also needs to be taken into consideration. The top of cell suspension is the area receiving UV irradiation and will rapidly lose efficacy within the solution. What was the depth of solution in the vial?

In accordance with the reviewer’s earlier comment concerning how our dose represents a sufficient but not minimal dose for SARS-CoV-2, this paragraph was changed to compare HSV-1 to published values (line 231).

3) Line 187 “certain downstream assays.” It is unclear at this point in the manuscript what these downstream assays would entail. Subsequent explanation of CLIP and other nucleic acid association assays are reasonable, but disconnected from this statement.

An example is now given to provide context (line 255).

4) Figure 2 title – “Pre-study experiment” is not informative and should be revised. The higher LOD is not sufficiently explained but presumptively reflects the 10 uL sample volume being dilute 1:100 for initial analysis? It is unclear if this "n" refers to the number of replicate experiments or samples evaluated at each UV inactivation dose.

We changed the title to “Mixing cells during irradiation increases inactivation efficiency” (line 184). N is now defined as “Each dose curve was performed with n=1” (line 189). An explanation of the different LoD was added to the methods section (line 94).

5) Supplemental Instructions line 23 – The last step should indicate the agent of disinfection and time, i.e. 70% ethanol for 5 minutes.

We changed the text to “Disinfect the outside of the tissue vials by wiping with Incidin Plus.“

Reviewer #2: Nice work. Please spell out short from in first line of Abstract. UV light can damage proteins. This may be something to keep in mind in future work. The UV dose you are using is quite high. It would be good if you could reduce it.

We have defined reverse transcription polymerase chain reaction (RT PCR) (line 27) and added an additional section where we discuss the used UV dose (line 316).

---

## [Editor Report · Decision Letter 1]

22 Aug 2022

A validated protocol to UV-inactivate SARS-CoV-2 and herpesvirus-infected cells

PONE-D-22-19938R1

Dear Dr. Bosse,

We’re pleased to inform you that your manuscript has been judged scientifically suitable for publication and will be formally accepted for publication once it meets all outstanding technical requirements.

Kind regards,

Etsuro Ito

Academic Editor

PLOS ONE

---

## [Editor Report · Acceptance letter]

12 Sep 2022

PONE-D-22-19938R1 

A validated protocol to UV-inactivate SARS-CoV-2 and herpesvirus-infected cells 

Dear Dr. Bosse:

I'm pleased to inform you that your manuscript has been deemed suitable for publication in PLOS ONE. Congratulations! Your manuscript is now with our production department. 

Kind regards, 

on behalf of

Prof. Etsuro Ito 

Academic Editor

PLOS ONE